# Longitudinal Analysis of Antimicrobial Resistance among *Enterococcus* Species Isolated from Australian Beef Cattle Faeces at Feedlot Entry and Exit

**DOI:** 10.3390/ani12192690

**Published:** 2022-10-06

**Authors:** Yohannes E. Messele, Mauida F. Hasoon, Darren J. Trott, Tania Veltman, Joe P. McMeniman, Stephen P. Kidd, Wai Y. Low, Kiro R. Petrovski

**Affiliations:** 1The Davies Livestock Research Centre, School of Animal and Veterinary Sciences, University of Adelaide, Adelaide, SA 5005, Australia; yohannes.messele@adelaide.edu.au (Y.E.M.); wai.low@adelaide.edu.au (W.Y.L.); 2The Australian Centre for Antimicrobial Resistance Ecology, University of Adelaide, Adelaide, SA 5005, Australia; mauida.alkhallawi@adelaide.edu.au (M.F.H.); darren.trott@adelaide.edu.au (D.J.T.); tania.veltman@adelaide.edu.au (T.V.); stephen.kidd@adelaide.edu.au (S.P.K.); 3Meat & Livestock Australia, Level 1, 40 Mount Street, North Sydney, NSW 2060, Australia; jmcmeniman@mla.com.au; 4Research Centre for Infectious Disease, School of Biological Sciences, University of Adelaide, Adelaide, SA 5005, Australia

**Keywords:** microbiology, surveillance, multidrug resistance, cattle

## Abstract

**Simple Summary:**

The emergence of multidrug-resistant (MDR) enterococci is a global public and animal health concern. Due to a fear of their spread to humans, the occurrence of antimicrobial-resistance (AMR) in enterococci in agricultural production creates controversy between the producers and consumers. The aim of this study was to characterize the prevalence of antimicrobial-resistant enterococci isolated from beef cattle in South Australia at two time points, the entry and the exit (at slaughter). Overall, the AMR prevalence remained largely static between the entry and the exit except for the increased resistance to nitrofurantoin and quinupristin/dalfopristin in the absence of antimicrobial selection pressure. Therefore, regular monitoring of AMR is essential to monitor resistance trends in beef cattle.

**Abstract:**

*Enterococcus faecium* are commensal bacteria inhabiting the gastrointestinal tract of animals and humans and an important cause of drug-resistant nosocomial infections. This longitudinal study aimed to determine whether changes in the antimicrobial resistance (AMR) phenotype and genotype occurred among *Enterococcus* spp. isolated from cattle rectal samples obtained at the entry to and exit from an Australian feedlot. The samples obtained at the feedlot induction yielded enterococci (104/150; 69.3%), speciated as *E. hirae* (90/104; 86.5%), *E. faecium* (9/104; 8.7%), *E. mundtii* (3/104; 2.9%), *E. durans*, and *E. casseliflavus* (1/104; 1.0% each). AMR was observed to lincomycin (63/104; 60.6%), daptomycin (26/104; 25.0%), nitrofurantoin (9/104; 8.7%), ciprofloxacin (7/104; 6.7%), tetracycline (5/104; 4.8%), tigecycline (4/104; 3.9%), and quinupristin/dalfopristin (3/104; 2.9%). From the rectal swab samples collected at the abattoir from the same animals (i.e., the feedlot exit), the enterococci recovery was significantly higher (144/150; 96.0%), with a marked shift in species distribution dominated by *E. faecium* (117/144; 81.3%). However, the prevalence of AMR to individual antimicrobials remained largely static between the entry and exit except for the increased resistance to nitrofurantoin (77/144; 53.5%) and quinupristin/dalfopristin (26/144; 18.1%). Overall, 13 AMR genes were observed among the 62 *E. faecium* isolates. These included *aac(6′)Ii*, *aac(6′)-Iid*, and *ant(6)-Ia* (aminoglycosides); *eatAv*, *lnu(G)*, *vat(E)*, *msr(C)*, and *erm(B)* (macrolides, lincosamides, and streptogramins); *efmA* (fluoroquinolones); and *tet(45)*, *tet(L)*, *tet(M)*, and *tet(S)* (tetracyclines). The results confirm the presence of fluoroquinolone- and streptogramin-resistant enterococci in cattle faeces at the feedlot entry in the absence of antimicrobial selection pressure. *E. faecium*, exhibiting increased nitrofurantoin resistance, became the dominant *Enterococcus* spp. during the feeding period.

## 1. Introduction

The rapidly growing demand for animal products has led to the intensification of animal production and associated with this is the increased use of antimicrobials to maintain animal health and welfare [1]. Antimicrobials have been widely used in livestock production systems, particularly in intensive feeding operations, for different purposes, including therapeutic, metaphylactic, and prophylactic uses for infectious disease treatment, control, and prevention [2]. The judicious use of antimicrobials as livestock treatments is vital in the face of few alternatives for specific diseases, while their overuse and/or misuse has led to the development of antimicrobial resistance (AMR). Antimicrobial treatments applied to beef cattle indiscriminately affect pathogenic bacteria present at the site of infection but also may impact the commensal microbiota of the gastrointestinal tract [3,4]. This may result in the elimination of susceptible microbial populations, thereby reducing competition from resistant bacteria, which may proliferate under antimicrobial selection pressure [5].

Enterococci are natural inhabitants of the intestinal tract of different organisms, including cattle [6]. They are resilient organisms capable of surviving in a broad range of temperatures and pH levels [7]. In human medicine, the emergence of AMR to Gram-positive and broad-spectrum antimicrobials among enterococcal species such as *Enterococcus faecium* has made the treatment of the infections caused by these opportunistic pathogens a real challenge for clinicians [8]. Resistance to antimicrobial agents can occur through intrinsic mechanisms found universally in the bacterial genome, in spontaneous mutations, or through the acquisition of new genetic material through recombination [9]. Enterococci are intrinsically resistant to a number of antimicrobial agents, including β-lactams and aminoglycosides [10]. Among the main enterococcal species, *E. faecalis* is most important to public health and is intrinsically resistant to clindamycin (a lincosamide), quinupristin (streptogramin B class), and dalfopristin (streptogramin A class) through activity conferred by the expression of the *lsa* gene [11]. They also have the capacity to acquire new mechanisms of AMR, either by mutation or genetic recombination via plasmids and transposons. In this way, enterococci have acquired AMR to many classes of antimicrobials, including glycopeptides (vancomycin), macrolides, quinolones, streptogramins, and tetracyclines [12].

The emergence of multidrug-resistant (MDR) strains such as vancomycin-resistant enterococci (VRE) in livestock production and human clinical settings has become a globally significant concern, although VRE are yet to be reported in Australian livestock species [13]. MDR enterococci are able to survive in the gastrointestinal tract and potentially become the dominant flora under antimicrobial pressure [8]. Furthermore, gene transfer occurs readily between closely related enterococcal species, but can also occur between different genera [14]. For example, enterococci may transmit vancomycin resistance to more pathogenic microorganisms such as methicillin-resistant *Staphylococcus aureus* in human patients [15].

The potential transfer of AMR from the enteric bacteria found in animals to humans (or vice versa) is a current global public health concern [16]. In previous studies, some genetic similarity was reported to exist between enterococci isolated from animals and those causing human infections [10,17,18]. However, more recent studies based on whole genome sequence analysis have found that many enterococci isolated from animals are unrelated to the strains causing infections in humans [19,20]. The prevalence of AMR among enterococci isolated from animals and humans varies between geographical location and patterns of antimicrobial use [21,22], and the identification of antimicrobial-resistant bacteria in agricultural settings may at times create issues for both producers and consumers [23]. Australia has strict registration and regulation of antimicrobial use and conducts periodic assessments on the status of antimicrobial-resistant bacteria in healthy livestock at slaughter [13,24,25]. Despite these restrictions and the periodic study, continuous follow-up is needed to understand the colonization dynamics of AMR surveillance indicators and zoonotic genera under various forms of production and antimicrobial selection pressure. Such a situation exists in cattle in Australia when, upon entry to a feedlot, they experience major changes in environment, diet, husbandry/management, and antibiotic treatments. Hence, the aim of this study was to determine whether changes in AMR status occur among *Enterococcus* spp. isolated from the same animals on entry into a beef feedlot farm and again at the slaughterhouse.

## 2. Materials and Methods

### 2.1. Study Population

This study was undertaken at a single commercial beef feedlot with an overall capacity of 17,000 head. For this study, a total of 150 cattle (initial BW = 405 kg) that originated from Location A (*n* = 82), Location B (*n* = 54), and Location C (*n* = 14) were used. The breeds of cattle were Angus, Hereford, Santa Gertrudis, Shorthorns, and their crosses. For the duration of the feedlot phase, all the cattle were housed in a single pen. The study cattle were fed a diet that consisted of variable proportions of tempered barley, lucerne hay, oaten hay, lupins, almond hull, and a molasses-based suspension supplement (containing macro-minerals, trace minerals, vitamins, and monensin). All the cattle were kept in a single pen for the 90-day feeding phase. During the study period, ceftiofur, oxytetracycline, and tulathromycin were the antimicrobials used to treat the sick cattle. In addition, the tetracycline-based product was also used for the metaphylaxis of cattle arriving from high-risk sources (e.g., cattle bought from saleyards); however, it was not used on the target pen as the study animals were not from high-risk areas. When an individual animal from the target pen exhibited early signs of illness, it was transferred to the hospital pen for the duration of the treatment period and then returned to the target pen when it had recovered. In this way, 13 of the 150 cattle (8.7%) were treated therapeutically with antimicrobials during the longitudinal study. The individual antimicrobial treatments included long-acting injections of either tulathromycin (*n* = 10), ceftiofur (*n* = 2), or oxytetracycline (*n* = 1). At the end of the feeding period, the cattle were transported to an abattoir 150 km from the feedlot. The cattle remained in lairage for less than 4 h, with access to clean water before slaughter.

### 2.2. Sample Collection

Individual-level faecal samples, with a mean weight of 15 g, were collected aseptically on entry by rectal grab with an aseptic gloved arm. The faecal samples were placed in a sterile screw-capped container on ice until arrival at the laboratory. In addition, faecal swab samples were collected at the abattoir using Ames transport media swabs (Copan, Italy) during post-evisceration by incision on the rectum 15–30 cm cranial to the anus. The swabs were placed on ice until arrival at the laboratory. All microbial analyses were initiated within 24 h of the sample collection.

### 2.3. Bacterial Isolation

Standard bacterial culture methods were used to isolate and identify enterococci from the samples [26]. Briefly, the faecal material was suspended into 7 mL of sterile 0.1% buffered peptone water in a falcon tube. The mixture was then spread onto a Slanetz and Bartley agar plate (Thermofisher Scientific, Adelaide, Australia) with a sterile cotton tip applicator and incubated at 37 ± 2 °C for 48 h. A red-, maroon-, or pink-coloured single colony was carefully chosen and subcultured onto sheep blood agar (Thermofisher Scientific, Australia). All suspected target colonies were confirmed to the species level by matrix-assisted laser desorption ionisation time-of-flight mass spectrometry (MALDI-TOF) (Bruker Daltonik GMBH, Germany) and stored at −80 °C in tryptone soy broth with 20% glycerol until future processing.

### 2.4. Phenotypic Determination of Antimicrobial Resistance

The antimicrobial minimum inhibitory concentrations (MICs) for all the Enterococcus spp. isolates were determined using the Sensititre automated system (Trek Diagnostic Systems, Thermofisher Scientific, Oxford, UK), following the Clinical and Laboratory Standards Institute guidelines [27] and the National Antimicrobial Resistance Monitoring System [28]. Susceptibilities were determined using the NARMS Gram-positive CMV3AGPF plate, which included chloramphenicol, ciprofloxacin, daptomycin, erythromycin, gentamycin, kanamycin, lincomycin, linezolid, nitrofurantoin, penicillin, quinupristin/dalfopristin, streptomycin, tetracycline, tigecycline, tylosin tartrate, and vancomycin. The reference strains used for quality control were *E. faecalis* ATCC 29212 and *S. aureus* ATCC 29213. The resistance breakpoints for each antimicrobial agent are shown in Table 1. All the isolates that yielded MICs above the CLSI susceptible breakpoint were reported as resistant.

### 2.5. Whole Genome Sequencing and Identification of Antimicrobial Resistance Genes

The *E. faecium* isolates found to be resistant to ciprofloxacin, daptomycin, erythromycin, quinupristin/dalfopristin, and/or tigecycline (*n* = 62) were further investigated for the presence of AMR genes by whole genome sequencing (WGS). Genomic DNA was extracted with a QIASymphony Virus/Pathogen DSP kit on a QIASymphony instrument according to the manufacturer’s instructions. WGS was performed with the NextSeq 550 platform and the NextSeq MID output (2 × 150 bp) paired-end sequencing kits. Library preparation was undertaken using Nextera XT Library preparation with Nextera XT indices. The reads were trimmed with the software Trimmomatic v0.38 to remove sequencing adapters and low-quality bases [29]. FASTQC v0.11.4 was used to check the quality of the raw and cleaned reads [30]. De novo genome assembly of the isolates was performed on cleaned reads using SPAdes v3.12.0 [31]. The assemblies were checked with Quast v4.5 for the number of contigs and contig N50 [32]. The AMR genes were predicted by searching the Antibiotic Resistance Genes Database (ARDB), the Comprehensive Antibiotic Resistance Database (CARD; https://card.mcmaster.ca) accessed on 23 January 2022 [33], the PointFinder database 1 February 2021) [34], and the ResFinder 4.0 EFSA 2021 database (24 May 2022) [35]. The default thresholds for the detection of antimicrobial resistance genes were used. The percent identity and coverage for ResFinder and PointFinder were 95% and 60%, respectively. The description of the result was based on the PointFinder and ResFinder output. The correlation between the resistance isolates and the resistance genes was also determined using BLASTn (https://blast.ncbi.nlm.nih.gov/Blast.cgi, accessed on 23 January 2022).

### 2.6. Statistical Analysis

The MIC distribution data for each antimicrobial were analysed using commercially available statistical analysis software (STATA version 15.0, Stata Corporation, College Station, TX, USA). The outcome of each MIC result was dichotomised into sensitive and resistant based on the predetermined cut-off value (Table 1). The isolates were categorised as multidrug-resistant if they exhibited resistance to one or more antimicrobials in three or more antimicrobial classes [36]. The frequency of resistance to each antimicrobial agent was described as rare: <0.1%; very low: 0.1% to 1.0%; low: >1.0% to 10.0%; moderate: >10.0% to 20.0%; high: >20.0% to 50.0%; very high: >50.0% to 70.0%; and extremely high: >70.0%, according to the European Food Safety Authority and the European Centre for Disease Prevention and Control [37].

## 3. Results

### 3.1. Prevalence of Antimicrobial Resistance at Entry into the Feedlot

Overall, 104/150 (69.3%) of the faecal samples obtained at entry to the feedlot yielded *Enterococcus* spp., with *Enterococcus hirae* being the predominant species recovered (90/104; 86.5%), followed by *E. faecium* 9/104; 8.7%), *Enterococcus mundtii* (3/104; 2.9%), *E. durans,* and *E. casseliflavus* (1/104; 1.0% each). Resistance was observed to lincomycin (63/104; 60.6%), daptomycin (26/104; 25.0%), nitrofurantoin (9/104; 8.7%), ciprofloxacin 7/104; 6.7%), tetracycline (5/104; 4.8%), tigecycline (4/104; 3.9%), and quinupristin/dalfopristin (3/104; 2.9%). All the isolates were sensitive to chloramphenicol, gentamycin, kanamycin, linezolid, penicillin, streptomycin, and vancomycin. The MIC distribution for each antimicrobial is shown in Table 2.

### 3.2. Prevalence of AMR at Exit from the Feedlot

Overall, 144/150 (96.0%) of the faecal samples obtained from the same animals at the exit from the feedlot (i.e., at the abattoir) yielded *Enterococcus* spp. However, *E. faecium* was the dominant isolate at this timepoint (117/144; 81.3%), followed by *E. hirae* (25/144; 17.4%), *E. durans*, and *E. mundtii* (1/144; 0.7% each). A high level of resistance was observed to lincomycin (121/144; 84.0%), followed by nitrofurantoin (77/144; 53.5%), daptomycin (33/144; 22.9%), quinupristin/dalfopristin (26/144; 18.1%), ciprofloxacin (11/144; 7.6%), and tetracycline (10/144; 6.9%). All the isolates were sensitive to chloramphenicol, gentamycin, linezolid, penicillin, and vancomycin (Table 3).

### 3.3. Antimicrobial Resistance Profiles

At the feedlot entry, 79/104 (76.0%) *Enterococcus* spp. isolates were resistant to at least one of the tested antimicrobial classes. These included 50/104 (48.1%) resistant to one, 18/104 (17.3%) resistant to two, 8/104 (7.7%) resistant to three, and 2/104 (1.9%) resistant to four antimicrobial classes. At the feedlot exit, 41/144 (28.5%) isolates were classified as MDR, while just 4/144 (2.8%) isolates were sensitive to all the tested antimicrobials (Table 4). Resistance to lincomycin, nitrofurantoin, and a variable third antimicrobial were the most frequently observed AMR profiles among the MDR *Enterococcus* spp. isolates obtained at the feedlot exit.

### 3.4. Changes in Antimicrobial Resistance Status Observed among Enterococcus faecium and E. hirae between Feedlot Entry and Exit

In this study, *Enterococcus faecium* and *E. hirae* were the two most frequently isolated *Enterococcus* spp. For each species, the frequency of resistance to each antimicrobial between entry and exit is shown in Figure 1 (Appendix A). Quinupristin/dalfopristin and ciprofloxacin resistance was only observed among the *E. faecium* isolates, with higher ciprofloxacin resistance frequency observed at the entry (7/9; 77.8%) compared to the exit (11/117; 9.4%). However, a much higher frequency of nitrofurantoin resistance was detected at the exit (*n* = 72/117; 61.5%). Overall, higher daptomycin resistance and non-sensitivity to tigecycline was observed in *E. hirae* compared to *E. faecium*. The prevalence of quinupristin/dalfopristin resistance and tigecycline non-sensitivity remained the same at both the entry and the exit. All the isolates were sensitive to vancomycin, chloramphenicol, gentamycin, linezolid, and penicillin.

The resistance profiles of the *E. faecium* and *E. hirae* isolates are shown in Appendix A. All the *E. faecium* isolates obtained at the entry and exit were resistant to at least one antimicrobial (Figure 2), while 26/90 (28.9%) of the *E. hirae* isolates obtained at the entry and 4/25 (16%) obtained at the exit were susceptible to all 16 tested antimicrobials. Among the *E. faecium* isolates, 3/9 (33.3%) obtained at the entry and 34/117 (29.1%) obtained at the exit were MDR. By comparison, among the *E. hirae* isolates, only 7/90 (7.8%) obtained at the entry and 6/25 (24%) obtained at the exit were MDR.

### 3.5. Antimicrobial Resistance Genes Identified among Enterococcus faecium

The antimicrobial-resistant *Enterococcus faecium* genomes were screened against the CARD and ResFinder databases for AMR genes (ARGs). Overall, 13 AMR genes were observed among 62 *E. faecium* isolates. These included *aac(6′)Ii*, *aac(6′)-Iid*, and *ant(6)-Ia* (aminoglycosides); *eatAv*, *lnu(G)*, *vat(E)*, *msr(C)*, and *erm(B)* (macrolides, lincosamides, and streptogramins); *efmA* (fluoroquinolones); and *tet(45), tet(L)*, *tet(M),* and *tet(S)* (tetracyclines) (Table 5). In addition, almost half the isolates (29/62; 46.7%) had a point mutation in the penicillin-binding protein (*pbp5*) gene that is responsible for engendering resistance to ampicillin.

Among the 62 *E. faecium* isolates selected for whole genome sequencing, the number of ARGs in the individual isolates ranged from one to eight, with 60/62 (96.8%) of the isolates carrying at least three ARGs (Appendix A). In most cases, a high level of agreement was observed between the resistance phenotype expressed by the isolate and the detection of a resistance-encoding ARG (Table 6). However, no resistance genotype was identified that could account for either the daptomycin resistance phenotype identified in 22/62 (35.5%) of the isolates or the nitrofurantoin resistance phenotype in 27/62 (43.5%) of the isolates.

## 4. Discussion

Enterococci have the potential to develop resistance to almost all the classes of antimicrobials of importance to human medicine [38]. Whilst both *E*. *faecalis* and *E*. *faecium* are associated with human infections, a higher proportion of VRE belong to *E. faecium* [39,40]. However, the improved understanding of AMR among enterococci would benefit from the inclusion of additional species in the AMR surveillance programmes [19]. This study focused on the AMR phenotypes and genotypes identified among enterococci isolated from cattle faeces collected from the same animals at the entry to and the exit from a beef feedlot in Southern Australia. The shift in specific *Enterococcus* spp. isolated, with a higher prevalence of *E. faecium* identified at the exit compared to the entry (which was dominated by *E. hirae*), was a noteworthy finding of the study. Second, among the enterococci species isolated, *E. faecium* more frequently expressed an MDR phenotype to several combinations of antimicrobials. This included both ciprofloxacin and quinupristin/dalfopristin resistance (detected only in *E. faecium* isolates), with the majority obtained from samples collected at the entry to the feedlot. Third, no ARGs were identified that could account for the moderate to high frequency of daptomycin resistance observed among the enterococci isolates, with *E. hirae* significantly higher compared to *E. faecium*. Fourth, no ARGs were identified that could account for the nitrofurantoin resistance among the *E. faecium* isolates, which increased markedly between the entry and exit samples.

The most prevalent *Enterococcus* spp. isolated at the feedlot entry was *E. hirae* (86.5%)*,* followed by *E. faecium* (8.7%), and *E. mundtii* 3 (2.9%). This result is in line with previous reports that indicated that *E. hirae* and *E. faecium* are frequent species detected in the faecal content of healthy animals [38,41]. Similarly, *E. hirae* was reported elsewhere as the most predominant species detected in beef cattle [19]. In the present study, a very high proportion of enterococci at the feedlot entry were resistant to lincomycin (60.6%) and daptomycin (25.0%), with lower proportions resistant to nitrofurantoin (8.7%), ciprofloxacin (6.7%), tetracycline (4.8%), tigecycline (non-susceptible 3.9%), and quinupristin/dalfopristin (2.9%). Multidrug resistance was commonly observed. Resistance to daptomycin was more likely to be present among the *E. hirae* isolates, whereas resistance to ciprofloxacin and quinupristin/dalfopristin was observed more frequently in *E. faecium*. These AMR trends have been reported in previous international studies. For example, in one Canadian study resistance varied between *E. faecium* and *E. hirae* for tetracycline (45% vs. 59%), nitrofurantoin (45% vs. 16%), macrolides (29% vs 33%), tigecycline (3% vs. 12%), and quinupristin/dalfopristin (3% vs. 1.4%) [19].

The present study confirmed that cattle arriving at the feedlot may already be colonised with *Enterococcus* spp. resistant to critically important antimicrobials that are not used in livestock in Australia. These antimicrobials include daptomycin, ciprofloxacin, nitrofurantoin, quinupristin/dalfopristin, and tigecycline (non-susceptibility), though it is fair to say that the ARGs conferring these resistant phenotypes were only identified in our *E. faecium* collection for ciprofloxacin (*efmA*) and quinupristin/dalfopristin (*eatAv, msr(C), vat(E)*). None of the classes these antimicrobials belong to is registered for use in livestock in Australia apart from virginiamycin, a streptogramin which can be strictly used only for the management of acute rumen acidosis [42]. As these antimicrobials are all used in human medicine, it is possible that they originate from background environmental sources at the point of origin of the feeder cattle. In this study, the effect of breed was tested and was found insignificant. Whilst AMR can be spread from humans to animals by transfer of the resistant bacteria through direct contact, further interrogation of the *E. faecium* genomes is required to determine their origins and transmissibility [43].

Unlike at the entry, *E. faecium* (81.3%) became the most predominant *Enterococcus* spp. identified at the exit, followed by *E. hirae* (17.4%). The change in diet from grass to a more concentrated energy/protein rich ration is the most likely reason for the observed change in species diversity, although age may also have been a factor in the altered faecal microbial community [44]. Similarly to our study findings, *E. faecium* isolates of animal origin have been found to be resistant to ciprofloxacin, tetracycline, and nitrofurantoin [45]. Ciprofloxacin resistance is more commonly detected in *E. faecium* compared to other enterococci [19,46]. The levels of resistance to daptomycin, erythromycin, lincomycin, and tetracycline in this study were also consistent with other Australian studies (abattoir surveys), which focused on both grazing and feedlot cattle [13,47], pigs [25], and poultry [20].

Antimicrobial resistance, particularly multi-resistance, is common among enterococci because of their ability to acquire ARGs [48]. In total, 41 isolates (29%) were MDR in the present study, with some isolates resistant to up to five antimicrobial classes. The emergence of new antimicrobial resistance in enterococci is likely associated with their ability to acquire new genetic elements through horizontal gene transfer (HGT) [49]. Additionally, innate resistance to some antimicrobials also must be considered [50]. In the present study, deeper interrogation of the genomes will be required to map ARGs to particular mobile genetic elements, the origin of which (human, animal, or environmental) will require further study.

Enterococci are naturally resistant to many classes of antimicrobials, such as aminoglycosides and β-lactams, and can also acquire resistance to other classes, including glycopeptides, quinolones, and tetracyclines [51]. In this study, the mutated form of the wildtype *eatA* ABC-F subfamily protein *eatAv* gene, which confers resistance to lincosamides, streptogramin A and pleuromutilins, was observed in 75.8% of the *E. faecium* isolates. The antimicrobial efflux pump *efmA* gene, important for the removal of macrolide and fluoroquinolone antimicrobials from the intracellular environment of bacterial cells, was observed in 66.7% of the ciprofloxacin-resistant *E. faecium* isolates. The prevalence of ciprofloxacin resistance among *E. faecium* isolates was higher in the much smaller number of isolates obtained at the entry compared to the exit samples. The dual nature of the resistance imparted by *efmA* likely explains the high prevalence of resistance in the absence of fluoroquinolone selection pressure, given that this antimicrobial class has never been registered for use in Australian food-producing animals. Ciprofloxacin resistance occurs through either the chromosomal mutation of DNA gyrase (*gyrA*) and topoisomerase IV (*ParC)* genes, the active efflux pump (*efmA*), target protection (*Qnr*-like determinants), or combinations thereof [52,53,54]. In the present study, only the efflux pump gene *efmA* was identified and both the *gyrA* and the *parC* genes did not contain mutations in their quinolone resistance determining regions, confirming that none of the isolates has developed point mutations under previous fluoroquinolone selection pressure [55].

Daptomycin resistance is reported to be linked with mutations of the genes encoding the cell envelope stress response (*LiaFSR* and *YycFGHIJ*) and the genes responsible for the metabolism of phospholipids (*gdpD* and *cls*) [56,57]. In this study, the WGS analysis revealed no mutation in these target genes that could account for the isolates with the MICs over the resistance breakpoint. Thus, at this point in time, the molecular mechanism of daptomycin resistance in enterococci is yet to be fully elucidated, and the relative impact of the use of other drug classes in the human vs. animal contexts on its distribution is completely unknown.

Interestingly, in the present study we found a high proportion of *E. faecium* isolates obtained at the feedlot exit (compared to the feedlot entry) that were resistant to nitrofurantoin, an antimicrobial used to treat urinary tract infection in humans [58] that has not been used in livestock worldwide since the early 1990s [59] and that, to the best of the authors’ knowledge, has never been used in livestock in Australia. High frequencies of nitrofurantoin resistance have also been reported in *E. faecium* isolated from feedlot cattle elsewhere (e.g., in Canada 45%) [19]. However, we hypothesise that nitrofurantoin resistance in this study may be yet to be elucidated or that, possibly, reverse zoonotic transfer has occurred. Nitrofurantoin resistance in human medicine occurs through the development of mutations in *nfsA* and/or *nfsB*, both of which encode oxygen insensitive nitroreductases [60]. In addition, the plasmid-mediated efflux genes, *oqxAB,* are associated with high-level nitrofurantoin resistance [61]. However, neither the mutation nor the efflux ARG were detected in the nitrofurantoin-resistant *E. faecium* isolates selected for sequencing in the present study. In a recent study of human *E. faecium* isolates obtained from urinary tract infections in China, the absence of the nitroreductase-encoding genes *ef0404* and *ef0648* was associated with high-level nitrofurantoin resistance [62]. However, a detailed scan of our *E. faecium* genomes found no correlation between gene absence and nitrofurantoin resistance (data not shown). These findings confirm that the mechanism of nitrofurantoin resistance in beef cattle *E. faecium* isolates remains not fully understood at this point in time, and further research is required to determine if it is chromosomally or mobile genetic element-encoded. Resistance outcomes for one antimicrobial class can be linked with resistance to other classes through co-selection [63].

Although the determination of the resistance phenotype and ARGs present in the commensal enterococci inhabiting healthy feedlot cattle at the entry to and exit from the feedlot has provided valuable insight, this study had some limitations. First, the study was conducted at a single beef feedlot in southern Australia, whereas many of Australia’s largest feedlots are distributed in the sub-tropical zones of Queensland and New South Wales. Second, we were unable to determine the effect of antimicrobial treatment on the development of AMR as only 13 cattle received curative treatment (mostly macrolides) during the 90-day feeding period. Third, faecal samples could not be obtained from the cattle at the feedlot immediately before transport to the abattoir; hence, the exit samples could only be obtained post-slaughter, and microbial population changes may have occurred during transport. Larger scale multi-site longitudinal studies are therefore recommended to fully investigate the bacterial AMR status in Australian feedlot cattle production systems. Future whole genome sequencing studies should also consider the associations between the AMR genes, plasmids, virulence factors, and genetic relatedness of the isolates.

## 5. Conclusions

In this study, major shifts in enterococci populations in the faeces of healthy feedlot cattle were detected between entry to and exit from the feedlot. These included the fact that *E. faecium* (including strains resistant to critically important antimicrobials) were isolated from only a few of the study cattle at the feedlot entry, whereas *E. faecium* was the predominant *Enterococcus* spp. isolated at the exit. The resistance among the enterococci was similar to or less than has been reported in international studies and was similar to the previously reported slaughter-based surveys for Australian cattle. The AMR phenotype and the possession of the corresponding ARGs were in agreement for the majority of cases but could not be established for daptomycin and nitrofurantoin resistance. Studies should now focus on a deeper analysis of the *E. faecium* genomes to determine their genetic relatedness to isolates obtained from human, livestock, and environmental sources.

## Figures and Tables

**Figure 1 animals-12-02690-f001:**
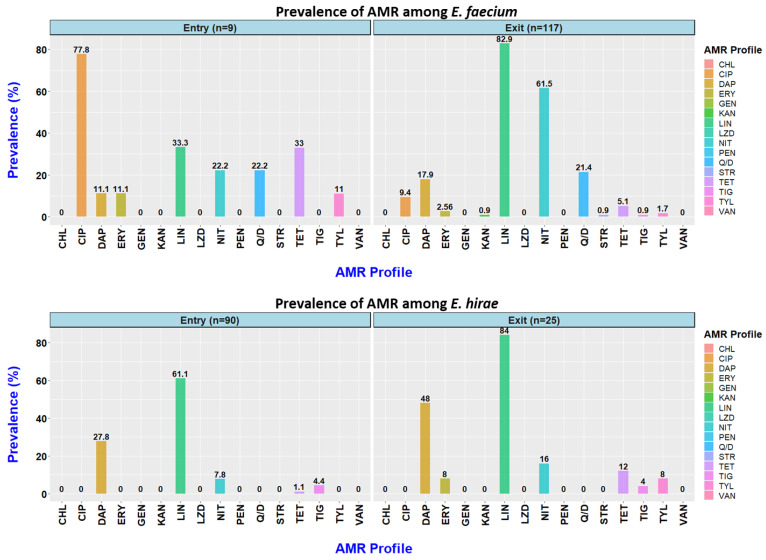
Prevalence of resistance to 16 antimicrobials observed among *Enterococcus faecium* and *Enterococcus hirae* isolated from faecal samples obtained at entry to (9 *E. faecium*; 90 *E. hirae*) and exit from (117 *E. faecium*; 25 *E. hirae*) an Australian feedlot.

**Figure 2 animals-12-02690-f002:**
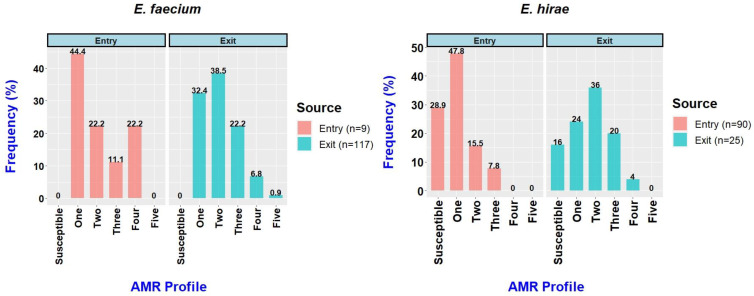
Frequency of *Enterococcus faecium* and *Enterococcus hirae* isolates from feedlot cattle faecal samples at entry and exit to the feedlot.

**Table 1 animals-12-02690-t001:** Dilution ranges and breakpoints used for antimicrobial susceptibility testing of Enterococcus isolates of faecal origin.

Antimicrobial Agent	Range	Breakpoints for Resistance
Chloramphenicol	2–32	≥32 ^a^
Ciprofloxacin	0.12–4	≥4 ^a^
Daptomycin	0.25–16	≥8 ^a^
Erythromycin	0.25–8	≥8 ^a^
Gentamicin	128–1024	≥512 ^b^
Kanamycin	128–1024	≥1024 ^b^
Lincomycin	1–8	≥8 ^b^
Linezolid	0.5–8	≥8 ^a^
Nitrofurantoin	2–64	>64 ^a^
Penicillin	0.25–16	≥16 ^a^
Streptomycin	512–2048	≥1024 ^b^
Quinupristin/Dalfopristin	0.5–32	≥4 ^a^
Tetracycline	1–32	≥16 ^a^
Tigecycline	0.015–0.5 ^c^	≥0.5 ^b^
Tylosine tartarte	0.25–32	≥32 ^b^
Vancomycin	0.25–32	≥32 ^a^

^a^ Breakpoint established by the Clinical and Laboratory Standards Institute; ^b^ breakpoint established by the National Antimicrobial Resistance Monitoring System; ^c^ only a breakpoint for susceptibility has been established [28].

**Table 2 animals-12-02690-t002:** Antimicrobial (*n* = 16) minimum inhibitory concentration (MIC) distribution data obtained for 104 *Enterococcus* spp. isolated from 150 rectal faecal samples at feedlot entry (*n* = 104). The vertical line within each MIC distribution indicates the resistance breakpoint (see Table 1) for each antimicrobial except for tigecycline, where it indicates the susceptible breakpoint.

Antimicrobial Class	Antimicrobial Agent	Resistance (95% CI)	Percentage of Isolates Yielding Each MIC Value (µg/mL)																
			0.015	0.03	0.06	0.12	0.25	0.5	1	2	4	8	16	32	64	128	256	512	1024
Aminoglycosides	Gentamycin	0.0 (0.00–3.56)														100.0			
	Kanamycin	0.0 (0.00–3.56)														98.1	1.0	1.0	
	Streptomycin	0.0 (0.00–3.56)																100.0	
Beta-lactam	Penicillin	0.0 (0.00–3.56)					23.1	20.2	31.7	21.2	3.9								
Fluoroquinolones	Ciprofloxacin	6.7 (3.24–13.45)					2.9	76.9	10.6	2.9	6.7								
Glycopeptides	Vancomycin	0.0 (0.00–3.56)					1.0	50	48.1		1.0								
Glycylcyclines	Tigecycline	4.8 (2.02–11.03)		14.4	43.3	29.8	7.7	4.8											
Lincosamide	Lincomycin	60.6 (50.91–69.48)							36.5	1.0	1.9	5.8	54.8						
Lipopeptides	Daptomycin	25.0 (17.62–34.19)					1.0		4.8	19.2	50.0	23.1	1.9						
Macrolides	Erythromycin	1.0 (0.14–6.51)					90.4	2.9	1.0	3.9	1.0		1.0						
	Tylosine tartrate	1.0 (0.14–6.51)					1.0	3.9	24.0	57.7	11.5	1.0			1.0				
Nitrofurantoins	Nitrofurantoin	8.65 (4.56–15.80)											2.9	40.4	48.1	8.7			
Oxazolidinones	Linezolid	0.0 (0.00–3.56)						1.0	3.9	94.2	1.0								
Phenicols	Chloramphenicol	0.0 (0.00–3.56)								1.0	92.3	6.7							
Streptogramins	Quinupristin/dalfopristin	2.9 (0.93–8.56)						38.5	17.3	41.4	1.9		1.0						
Tetracycline	Tetracycline	3.9 (1.45–9.80)							96.2					1.0	2.9				

**Table 3 animals-12-02690-t003:** Antimicrobial (*n* = 16) minimum inhibitory concentration (MIC) distribution data obtained for 144 *Enterococcus* spp. isolated from 150 rectal faecal samples obtained at feedlot exit. The vertical line within each MIC distribution indicates the resistance breakpoint (see Table 1) for each antimicrobial except for tigecycline, where it indicates the susceptible breakpoint.

Antimicrobial Class	Antimicrobial Agent	Resistance (95% CI)	Percentage of Isolates Yielding Each MIC Value (µg/mL)																
			0.015	0.03	0.06	0.12	0.25	0.5	1	2	4	8	16	32	64	128	256	512	1024
Aminoglycosides	Gentamycin	0.0 (0.00–2.60)														100.0			
	Kanamycin	0.7 (0.10–4.76)														70.1	25.7	3.5	0.7
	Streptomycin	0.7 (0.10–4.76)																99.3	0.7
Beta-lactam	Penicillin	0.0 (0.00–2.60)					8.3	12.5	11.1	23.6	43.8	0.7							
Fluoroquinolones	Ciprofloxacin	7.6 (4.28–13.27)					1.4	13.2	38.2	39.6	7.6								
Glycopeptides	Vancomycin	0.0 (0.00–2.60)						60.4	34.0	4.2	1.4								
Glycylcyclines	Tigecycline	1.4 (0.35–5.38)		0.7	3.5	52.1	38.2	4.2		1.4									
Lincosamide	Lincomycin	84.0 (77.11–89.15)							13.2	2.8		1.4	82.6						
Lipopeptides	Daptomycin	22.9 (16.77–30.48)							1.4	12.5	63.2	22.2	0.7						
Macrolides	Erythromycin	4.2 (1.88–8.96)					60.4	1.4	3.5	22.2	8.3	1.4	2.8						
	Tylosine tartrate	3.5 (1.45–8.07)							0.7	25.7	24.3	45.8			3.5				
Nitrofurantoins	Nitrofurantoin	53.5 (45.30–61.46)												3.5	43.1	53.5			
Oxazolidinones	Linezolid	0.0 (0.00–2.60)							0.7	97.2	2.1								
Phenicols	Chloramphenicol	0.0 (0.00–2.60)									10.4	89.6							
Streptogramins	Quinupristin/dalfopristin	18.1 (12.59–25.20)						13.2	4.2	64.6	17.4	0.7							
Tetracycline	Tetracycline	6.9 (3.78–12.43)							93.1					0.7	6.3				

**Table 4 animals-12-02690-t004:** Antimicrobial resistance profiles identified among *Enterococcus* spp. isolated from rectal faecal samples obtained at entry and exit from an Australian feedlot.

Antimicrobial Classes	Total No. of Isolates (%)		Resistance Pattern (No. of Isolates)	
	Entry (104)	Exit (144)	Entry	Exit
All susceptible	26 (25.00)	4 (2.78)	26	4
1	50 (48.08)	44 (30.56)	LIN (38)	LIN (38)
			DAP (6)	NIT (6)
			CIP (3)	
			TGC (2)	
			TET (1)	
2	18 (17.31)	55 (38.19)	DAP-LIN (11)	LIN-NIT (28)
			CIP-NIT (1)	LIN-Q/D (9)
			CIP-TET (1)	CIP-LIN (1)
			DAP-NIT (1)	DAP-NIT (7)
			LIN-TIG (3)	DAP-LIN (7)
			LIN-Q/D (1)	CIP-NIT (1)
				LIN-TET (1)
				ERY-LIN-TYL (1)
3	8 (7.69)	31 (21.53)	DAP-LIN-NIT (6)	LIN-NIT-Q/D (10)
			CIP-LIN-NIT (1)	CIP-LIN-NIT (3)
			DAP-LIN-TET (1)	CIP-DAP-NIT (3)
				CIP-NIT-TIG (1)
				DAP-LIN-TET (2)
				DAP-LIN-Q/D (1)
				DAP-LIN-NIT (8)
				NIT-STR-TET (1)
				LIN-NIT-TET (1)
				ERY-LIN-TIG-TYL (1)
4	2 (1.92)	9 (6.25)	CIP-DAP-LIN-Q/D (1)	DAP-LIN-NIT-TET (1)
			ERY-LIN-Q/D-TET-TYL (1)	KAN-LIN-NIT-Q/D (2)
				DAP-ERY-LIN-NIT (1)
				CIP-DAP-LIN-NIT (1)
				LIN-NIT-Q/D-TET (1)
				ERY-LIN-Q/D-TET-TYL (2)
				ERY-LIN-NIT-TET-TYL (1)
5		1 (0.69)		CIP-DAP-LIN-NIT-Q/D (1)
Non-MDR	68 (65.4)	99 (68.8)		
MDR	11 (10.6)	41 (28.5)		
Resistance	79 (76.0)	140 (96.3)		

CIP, Ciprofloxacin; DAP, Daptomycin; ERY, Erythromycin; KAN, Kanamycin; LIN, Lincomycin; NIT, Nitrofurantoin; Q/D, Quinupristin/dalfopristin; STR, Streptomycin; TET, Tetracycline; TIG, Tigecycline; TYL, Tylosine tartrate.

**Table 5 animals-12-02690-t005:** The frequency of antimicrobial resistance genes found in 62 *E. faecium* isolates selected for whole genome sequencing.

Antimicrobial Class	Resistance Phenotype	Resistance Gene	Number of Isolates (*n* = 62)
Aminoglycosides	GEN	*aac(6′)-Ii*	59 (95.2)
Aminoglycosides	AMK	*aac(6′)-Iid*	2 (3.2)
Aminoglycosides	STR	*ant(6)-Ia*	1 (1.6)
β-lactam	AMP	*pbp5*	29 (46.8)
LsaP (lincosamides, streptogramin As and pleuromutilins)	Q/D, LIN	*eatAv*	47 (75.8)
Lincosamide	LIN	*lnu(G)*	2 (3.2)
Streptogramin	VIR, Q/D	*vat -E*	1 (1.6)
Macrolide, streptogramin	ERY, Q/D, VIR	*msr(C)*	59 (95.2)
MLS (macrolide, lincosamide, streptogramin)	ERY, LIN, Q/D	*erm(B)*	3 (4.8)
Macrolides, fluoroquinolones	CIP	*efmA*	21 (33.9)
Tetracyclines	TET	*tet(M)*	3 (4.8)
Tetracyclines	TET	*tet(L)*	2(3.2)
Tetracyclines	TET	*tet(45)*	2(3.2)
Tetracyclines	TET	*tet(S)*	2(3.2)

**Table 6 animals-12-02690-t006:** Agreement between phenotypic and genotypic resistance among the 62 *E. faecium* isolates subjected to whole genome sequencing.

Antimicrobial Class	AMR Isolates (%)	Resistance Gene Observed (%)	Agreement (%)
Aminoglycosides	Kanamycin (*n* = 1; 1.6)	*aac(6′)-Ii* (*n* = 1;1.6)	100
	Streptomycin (*n* = 0)	*ant(6)-Ia* (*n* = 1; 1.6)	0
Fluoroquinolones	Ciprofloxacin (*n* = 18; 29.0)	*efmA* (*n* = 12; 19.3)	66.7
Lipopeptides	Daptomycin (*n* = 22; 35.5)		0
Lincosamide	Lincomycin (*n* = 39; 62.9)	*eatAv* (*n* = 38; 61.3)	97.4
		*erm(B)* (*n* = 3; 4.8)	
		*lnu(G)* (*n* = 2; 3.2)	
Macrolides	Erythromycin (*n* = 4; 6.4)	*msr(C)* (*n* = 4; 6.4)	100
		*erm(B)* (*n* = 3; 4.8)	
	Tylosin tartrate (*n* = 3; 4.8)	*erm(B)* (*n* = 3; 4.8)	100
Nitrofurantoin	Nitrofurantoin (*n* = 27; 43.5)		0
Streptogramins	Quinupristin/dalfopristin (*n* = 27; 43.5)	*eatAv* (*n* = 26; 41.9)	96.3
		*msr(C)* (*n* = 26; 41.9)	
		*Vat(E)* (*n* = 1; 1.6)	
Tetracycline	Tetracycline (*n* = 5; 8.1)	*tet(M)* (*n* = 3; 4.8)	100
		*tet(L)* (*n* = 2; 3.2)	
		*tet(S)* (*n* = 2; 3.2)	
		*tet(45)* (*n* = 2; 3.2)	

## Data Availability

All isolate WGS reads are available in the SRA under BioProject PRJNA879912.

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
