# Peer review of "Longitudinal Analysis of Antimicrobial Resistance among Enterococcus Species Isolated from Australian Beef Cattle Faeces at Feedlot Entry and Exit"

_animals, 2022, doi:10.3390/ani12192690_

Round 1
Reviewer 1 Report
In the submitted paper, authors investigated the changes in antimicrobial resistance phenotype and genotype in Enterococcus spp. that were isolated from cattle rectal samples at two points (entry to feedlot and also at slaughter house). In this, 150 animals from different cattle breeds (Angus, Hereford, Santa Gertrudis, Shorthorns and crosses) were included. The study is well designed with clear objectives. However authors did not mention the effect of breed of cattle on the frequency/resistance of Enterococci as intestinal microbiota has been believed to be influenced by the breed of animals. Addition of this information may enhance the interest of readers.
Author Response
Please see the attachement

Reviewer 2 Report
Dear Dr. Messele,
Your manuscript describes an very interesting longitudinal analysis of antimicrobial resistance in cattle using enterococci as target specie.
Overall, your manuscript is well structure and provides relevant information for the general reader, shedding light about the population dynamics of AMR in enterococci sourced from cattle. The presentation of the results and discussion is broad, providing and in-depth analysis of the data and comparison with the current literature.
Probably the main drawback of the manuscript can be found in the lack of molecular epidemiology data that could better explain, in combination with the entry and exit AMR results, the potential source of those resistant microorganisms. As the authors rightly comment in the conclusions, this is a much needed area to develop in combination with the information presented by the authors.
Please find below minor comments that should be modified before acceptance:
Line 15: delete “an”
Line 18: Change over all for overall
Line 61: Italicize Enterococcus faecium
Line 67: Italicize E. faecalis
Line 81: Italicize Staphylococcus aureus
Line 99: Italicize Enterococcus
Line 114: Change “used to metaphylaxis” for “used for the metaphylaxis of”
Line 126: Add “on entry” by rectal…
Line 128: Change “from the exit” for “at abattoir”
Line 161: Italicize E. faecium
Line 175-176: Change “We used the default thresholds for the detection of 175 antimicrobial resistance genes” for “the default thresholds for the detection of 175 antimicrobial resistance genes were used”
Page 7, 10 and 11 (no line number provided): Italicize Enterococcus spp., E. faecium
In conclusions (line 414) change predominate for predominant
Reviewer 3 Report
This is a very nice, well written and well executed study investigating Enterococcus spp., in beef cattle at entry to a feedlot, and at exit from it. The study (as is always the case) has a few limitations but the authors point these out in the discussion. The data is well researched and presented with some good figures which aid the reader in understanding the study. Good to see all the sequences uploaded for others to access.
I have made a few very minor comments below- which most refer to the need to have bacterial names written in italics.
But thank you for making a reviewers life easy. I thoroughly enjoyed reading the manuscript and hope to see it published soon.
Line 50- wrong reference style, please amend
Line 61- bacterial names needs to be in italics. And 67, and 81, and 99, and 161. And all section 3.2, 3.3, 3.4
Line 94- there is a need to understand … (reword)
Line 115 – not used on the target pen …. (reword)
Discussion line 107- comma after However rather than a full stop
Conclusion- this is more a summary than a conclusion, please reword
